# Retinoic Acid Sensitivity of Triple-Negative Breast Cancer Cells Characterized by Constitutive Activation of the notch1 Pathway: The Role of Rarβ

**DOI:** 10.3390/cancers12103027

**Published:** 2020-10-18

**Authors:** Gabriela Paroni, Adriana Zanetti, Maria Monica Barzago, Mami Kurosaki, Luca Guarrera, Maddalena Fratelli, Martina Troiani, Paolo Ubezio, Marco Bolis, Arianna Vallerga, Federica Biancardi, Mineko Terao, Enrico Garattini

**Affiliations:** 1Laboratory of Molecular Biology, Istituto di Ricerche Farmacologiche Mario Negri IRCCS, via Mario Negri 2, 20156 Milano, Italy; gabriela.paroni@marionegri.it (G.P.); adriana.zanetti@marionegri.it (A.Z.); mariamonica.barzago@marionegri.it (M.M.B.); mami.kurosaki@marionegri.it (M.K.); luca.guarrera@guest.marionegri.it (L.G.); maddalena.fratelli@marionegri.it (M.F.); martina.troiani@ior.usi.ch (M.T.); marco.bolis@marionegri.it (M.B.); arianna.vallerga@guest.marionegri.it (A.V.); federica.biancardi@unimi.it (F.B.); mineko.terao@marionegri.it (M.T.); 2Department of Oncology, Istituto di Ricerche Farmacologiche Mario Negri IRCCS, via Mario Negri 2, 20156 Milano, Italy; paolo.ubezio@marionegri.it; 3Functional Cancer Genomics Laboratory, Institute of Oncology Research, USI, University of Southern Switzerland, 6500 Bellinzona, Switzerland

**Keywords:** retinoic acid, breast cancer, triple-negative, Notch1, RARbeta

## Abstract

**Simple Summary:**

We provide experimental evidence that the rare subgroup of triple-negative breast cancer characterized by constitutive activation of the NOTCH1 signaling pathway is sensitive to the anti-tumor action of all-trans retinoic acid, the active metabolite of vitamin A. In this tumor context, all-trans retinoic acid exerts not only an effective action on its own, but it also stimulates the inhibitory activity of γ-secretase inhibitors, a series of therapeutic agents targeting NOTCH1, on cancer cell growth. From a basic and mechanistic standpoint, an important result of the study regards the specific involvement of the retinoid receptor RARβ in the anti-tumor action exerted by all-trans retinoic acid in sensitive triple-negative breast cancer cells. From an applicative point of view the study represents the basis for the design of clinical trials on the efficacy of combinations between all-trans retinoic acid and γ-secretase inhibitors in the treatment of patients affected by a specific subtype of triple-negative breast cancer.

**Abstract:**

Triple-negative breast cancer (*TNBC*) is a heterogeneous disease that lacks effective therapeutic options. In this study, we profile eighteen *TNBC* cell lines for their sensitivity to the anti-proliferative action of all-trans retinoic acid (ATRA). The only three cell lines (*HCC-1599*, *MB-157* and *MDA-MB-157*) endowed with ATRA-sensitivity are characterized by genetic aberrations of the *NOTCH1*-gene, causing constitutive activation of the NOTCH1 γ-secretase product, N1ICD. N1ICD renders *HCC-1599*, *MB-157* and *MDA-MB-157* cells sensitive not only to ATRA, but also to γ-secretase inhibitors (DAPT; PF-03084014). Combinations of ATRA and γ-secretase inhibitors produce additive/synergistic effects in vitro and in vivo. RNA-sequencing studies of *HCC-1599* and *MB-157* cells exposed to ATRA and DAPT and ATRA+DAPT demonstrate that the two compounds act on common gene sets, some of which belong to the NOTCH1 pathway. ATRA inhibits the growth of *HCC-1599*, *MB-157* and *MDA-MB-157* cells via RARα, which up-regulates several retinoid target-genes, including RARβ. RARβ is a key determinant of ATRA anti-proliferative activity, as its silencing suppresses the effects exerted by the retinoid. In conclusion, we demonstrate that ATRA exerts a significant anti-tumor action only in *TNBC* cells showing constitutive NOTCH1 activation. Our results support the design of clinical trials involving combinations between ATRA and γ-secretase inhibitors for the treatment of this *TNBC* subtype.

## 1. Introduction

Triple-negative breast cancer (*TNBC*) accounts for 15–20% of all mammary tumors, and it is characterized by estrogen-receptor (ER), progesterone-receptor and HER2-receptor negativity. Even if *TNBC* cells share common features such as a high proliferation index and a basal-like gene expression signature, this tumor type is very heterogeneous and lacks effective therapeutic strategies [1,2]. NOTCH1 is a transmembrane receptor and its constitutive activation is observed in approximately 3% of all *TNBC* cases [3,4]. Normally, NOTCH1 activation requires binding to a membrane tethered ligand on neighboring cells, which triggers a series of proteolytic events [5,6]. The final γ-secretase-dependent cleavage of NOTCH1 causes the release and nuclear translocation of the receptor intracellular domain (N1ICD), which is part of an active transcriptional complex controlling the expression of various target genes [7,8]. Among the known target genes, members of the HES and HEY families, CyclinD1 and cMyc stand out [3]. Some of these genes, with particular reference to cMyc, are involved in the proliferative effects induced by the activation of the NOTCH pathway in certain types of leukemia and solid tumors. All this supports the development of strategies based on NOTCH targeting agents, with particular reference to γ-secretase inhibitors, for the treatment of *TNBC* cases characterized by constitutive NOTCH1 activation [9,10]. However, the active dosages of γ-secretase inhibitors are characterized by systemic toxicity [11], supporting the necessity of identifying pharmacological agents boosting the activity and reducing the toxicity of these compounds.

All-trans retinoic-acid (ATRA) is the active metabolite of vitamin A and a non-conventional anti-tumor agent endowed with cyto-differentiating properties [12,13]. In combination with chemotherapy or arsenic trioxide, ATRA is used in the treatment of acute promyelocytic leukemia with outstanding results, inducing long-term remission in the majority of patients [14]. The therapeutic activity observed in this type of acute leukemia has raised interest in the use of ATRA and derived synthetic retinoids for the personalized management of solid tumors, including breast cancer [15]. In this last context, a substantial number of pre-clinical in vitro and in vivo results indicate that ATRA is a promising agent in the treatment/chemo-prevention of mammary tumors [12,16]. Recently, we presented data supporting the idea that the majority of luminal breast cancers are sensitive to the anti-tumor action of ATRA [17,18]. In contrast, only a small fraction of basal or *TNBC* tumors are likely to be responsive to the retinoid. In breast cancer cells, the anti-tumor action of ATRA is predominantly due to a growth-inhibitory effect [17]. However, we recently demonstrated that challenge of mammary tumor cells with the retinoid reactivates endogenous retroviruses causing a *viral-mimicry* response [19]. The process may be at least partially the consequence of epigenetic effects, including perturbations in the DNA methylation process [20,21]. Activation of *viral-mimicry* may have significant therapeutic ramifications, as the process results in interferon-dependent immune responses that are likely to sensitize the neoplastic cell to immune-checkpoint inhibitors and other immune-therapeutics. The biological action of ATRA is generally mediated by the activation of RARs and RXRs, which are members of the nuclear receptor family [12,22]. Nuclear receptors are ligand-activated transcription factors which control the activity of numerous target genes. ATRA is a pan-RAR agonist, activating the RARα, RARβ and RARγ retinoid receptors with equal efficiency. The anti-proliferative effect exerted by ATRA in sensitive breast cancer cell lines seems to be predominantly due to ligand-dependent activation of RARα [17].

In the present study, we demonstrate that ATRA exerts a significant anti-tumor action in *TNBC* cells characterized by constitutive NOTCH1 activation. In addition, we show that ATRA enhances the anti-tumor activity of γ-secretase inhibitors in a synergistic manner using cellular models of this *TNBC* subtype. Finally, we provide evidence that the RARα-dependent induction of RARβ plays an important role in the anti-proliferative action exerted by ATRA in sensitive *TNBC* cells.

## 2. Results

### 2.1. ATRA Sensitivity of TNBC Cell lines Characterized by N1ICD Expression

The *ATRA-score* experimental index, which we developed, is a reliable parameter for the assessment and prediction of the anti-proliferative action exerted by ATRA in breast cancer and other types of neoplastic diseases [17,18]. The low *ATRA-score* values determined in most *TNBC* cell lines indicate that this cell type is generally resistant to ATRA (Figure 1A). Indeed, only three (*HCC-1599*, *MB-157* and *MDA-MB157*) of the eighteen *TNBC* cell lines considered are endowed with high *ATRA-score* values and respond to ATRA. Two additional *TNBC* cell lines (*HCC-1806* and *HCC-70*) present with barely measurable *ATRA-score*s, while all the other ones show undetectable values. Remarkably, *HCC-1599*, *MB-157* and *MDA-MB157* cells are reported to be characterized by internal-deletions of the *NOTCH1* gene [4,23,24,25], which causes constitutive expression of N1ICD, the γ-secretase cleavage product and transcriptionally active form of NOTCH1 [26]. Consistent with this, *HCC-1599*, *MB-157* and *MDA-MB157* are the only *TNBC* cell lines expressing significant amounts of N1ICD (Figure 1A).

To verify the presence of *NOTCH1* gene alterations in *HCC-1599*, *MB-157* and *MDA-MB157* cells, we performed RNA-sequencing (RNA-seq) experiments. The results demonstrate an exon expression imbalance of the *NOTCH1* gene, which indicates the presence of an internal deletion (Figure 1B). In *HCC-1599* cells, the deletion involves exons 3–27, it eliminates the NRR (Negative Regulatory Region) from the primary NOTCH1 gene product [23] and it results in a gain-of-function phenotype [4,23,24,27,28]. In *MB-157* and *MDA-MB157* cells, we do not detect the product of the *SEC16A-NOTCH1* gene-fusion reported by Stoeck et al. [24], although the two cell lines show a deletion of *NOTCH1* exons 2-27. Sequencing of the PCR products (exon-1/exon-28 primers) obtained from the *HCC-1599*, *MB-157* and *MDA-MB157* RNA confirms the presence/position of the *NOTCH1* gene internal deletions (Figure 1C,D).

### 2.2. NOTCH1 Role in the Growth of ATRA-Sensitive TNBC Cell lines

To investigate whether N1ICD activation contributes to the proliferation of *HCC-1599*, *MB-157* and *MDA-MB157* cells, we conducted studies with the γ-secretase and NOTCH inhibitor, DAPT [N-(N-(3,5-difluorophenacetyl)-L-alanyl)-S-phenylglycine-t-butyl-ester]. First, we transfected *HCC-1599*, *MB-157*, *MDA-MB157* as well as three other *TNBC* and retinoid-insensitive cell lines (*MDA-MB231*; *HCC-38*; *MDA-MB436*) with a reporter construct, containing a luciferase cDNA driven by the minimal promoter of the NOTCH1 target-gene, *HES1* [29,30]. Consistent with constitutive NOTCH1 activation, *HCC-1599*, *MB-157* and *MDA-MB157* cells show high basal levels of luciferase activity, which are reduced by DAPT (Figure 2A). In contrast, vehicle- or DAPT-treated *MDA-MB231*, *MDA-MB436* and *HCC-38* cells are devoid of significant luciferase activity. In *HCC-1599*, *MB-157* and *MDA-MB157* cells, DAPT-dependent suppression of luciferase activity is due to the expected inhibition of NOTCH1 cleavage into N1ICD (Figure 2B). Subsequently, we evaluated the anti-proliferative action of DAPT (1 μM, 9 days). DAPT reduces the proliferation/survival of *HCC-1599*, *MB-157* and *MDA-MB157* cells, while it does not affect the growth of *MDA-MB231*, *MDA-MB436* and *HCC-38* cells (Figure 2C). In *HCC-1599*, *MB-157* and *MDA-MB157* cells, the action of DAPT is time- and concentration-dependent (Figure 2D) and it is replicated with PF-03084014 [31,32,33], another γ-secretase inhibitor (Figure 2E).

### 2.3. Cross-Resistance of DAPT-Resistant Cells to ATRA

A comparison of the dose-response and time-dependent curves obtained in *HCC-1599*, *MB-157* and *MDA-MB157* cells with ATRA (Appendix A) and DAPT/PF-03084014 (Figure 2D,E) indicates that the more a cell line is sensitive to the retinoid, the more it responds to γ-secretase inhibitors. To support the idea, we developed *MB-157* cells characterized by induced DAPT resistance. Long-term culturing of *MB-157* cells in the presence of DAPT (1 µM) resulted in the isolation of two independent and DAPT-resistant cell lines of clonal origin (*MB-157RCL7A* and *MB-157RCL15A*). Relative to the parental *MB-157* counterpart, *MB-157RCL7A* and *MB-157RCL15A* cells show resistance not only to DAPT, but also to ATRA (Figure 3A). Cross-resistance is specific to ATRA, as no difference in sensitivity of the three cell lines to other chemotherapeutics, such as VP16 (etoposide-phosphate), is observed (Figure 3B). This cross-resistance to DAPT and ATRA supports NOTCH1 involvement in the anti-proliferative responses to the retinoid.

### 2.4. Synergistic Anti-Tumor Effects of ATRA and γ-Secretase Inhibitors

We studied the growth-inhibitory effects of ATRA, DAPT and ATRA+DAPT combinations in retinoid-sensitive *HCC-1599*, *MB-157* and *MDA-MB157* cells as well as retinoid-resistant *MDA-MB231*, *MDA-MB436* and *HCC-38* cells (Figure 3C). ATRA causes a time-dependent reduction in the growth of *HCC-1599*, *MB-157* and *MDA-MB157* cells. The anti-proliferative effect of DAPT is obvious in *HCC-1599* and *MB-157*, while it is less evident in *MDA-MB157* cells. In *MB-157* cells, ATRA+DAPT is significantly more active than ATRA or DAPT alone. A similar effect is likely to occur in *HCC-1599* cells (Figure 3C), although the strong anti-proliferative action of ATRA and DAPT alone masks the interaction. As for *MDA-MB157* cells, they are only mildly sensitive to DAPT (see also Figure 2C) and the combination of ATRA+DAPT is no more effective than ATRA alone. As for *MDA-MB231*, *MDA-MB436* and *HCC-38*, the lack of constitutive NOTCH1 activation renders the 3 cell lines similarly resistant to DAPT, ATRA and ATRA+DAPT. In *HCC-1599* and *MB-157* cells, we evaluated the additive/synergistic nature of the potentiating effects exerted by ATRA+DAPT. Isobologram analysis [34,35] of the dose-response data shows synergistic interactions between ATRA and DAPT in both cell lines (Figure 3D). In *HCC-1599* cells, the synergism is confirmed with combinations of ATRA and PF-03084014 (Appendix A).

To establish whether the therapeutic interaction between the retinoid and γ-secretase inhibitors can be replicated in vivo, we performed experiments with xenografts of *HCC-1599* cells, which have been shown to be responsive to the retinoid [17] and PF-03084014 [31] in previous studies. Tumor-bearing animals were treated with vehicle, ATRA, PF-03084014 or ATRA + PF-03084014 for 18 days. The volume of the tumors was determined up to 15 days following treatment cessation (Figure 4A). As expected, administration of ATRA and PF-03084014 alone reduces the growth of *HCC-1599* tumors. In addition, the tumor-growth curves indicate that ATRA + PF-03084014 is more effective than each compound alone. The phenomenon is particularly evident following discontinuation of the treatment. Indeed, ATRA + PF-03084014 causes a remarkable reduction of the tumor re-growth observed following the end of the treatment in animals administered ATRA and PF-03084014 alone. The efficacy of the ATRA + PF-03084014 combination was further evaluated using two other parameters, i.e., “*Percentage-Growth-Inhibition*” (%GI) and “*Absolute-Growth-Delay*” (*AGD*). The *%GI* value is indicative of short-term anti-tumor effects, while *AGD* defines the long-term delay of tumor regrowth. ATRA + PF-03084014 causes a more significant reduction of the %GI value than ATRA or PF-03084014 alone (Figure 4B). In addition, and consistent with an increased long-term efficacy of the combination, the AGD value is higher following treatment with ATRA + PF-03084014 than ATRA or PF-03084014 alone (Figure 4C). In our experimental conditions, treatment of the tumor bearing animals with ATRA, PF-03084014 or ATRA + PF-03084014 do not cause significant effects on the body weight of xenografted animals (Appendix A). Consistent with what is observed in the cultures of *HCC-1599* cells, the tumor tissue of vehicle treated control animals expresses only the N1ICD form of the NOTCH1 receptor (see Figure 1A). In addition, and as expected, the antitumor effects observed with PF-03084014 are associated with a substantial reduction in the levels of the N1ICD protein synthesized by the breast cancer cells [32] (Figure 4D). Interestingly, a similar reduction in the amounts of the N1ICD protein is also observed following administration of ATRA. Finally, the N1ICD protein is undetectable in tissue samples obtained from animals treated with the ATRA + PF-03084014 combination.

### 2.5. Effects of ATRA on NOTCH1 Cleavage into N1ICD

The *NOTCH1* gene internal deletion observed in *HCC-1599*, *MDA-MB157* and *MB-157* cells results in the expression of an amino-terminal deleted transmembrane precursor protein which is constitutively cleaved into N1ICD by γ-secretase (Figure 1A) [24,26]. In the three cell lines, the transmembrane precursor is detected only upon exposure to a γ-secretase inhibitor like DAPT. Thus, we evaluated whether ATRA, DAPT and ATRA+DAPT perturb the levels of N1ICD constitutively expressed in *HCC-1599*, *MDA-MB157* and *MB-157* cells. As expected, in all these cell lines, DAPT suppresses N1ICD expression and causes the appearance of the NOTCH1 transmembrane precursor protein, as a consequence of γ-secretase inhibition (Figure 5A). In contrast, ATRA reduces the amounts of N1ICD only in *HCC-1599* and *MDA-MB157* cells. Consistent with this, *HCC-1599* and *MDA-MB157* are the sole cell lines showing a reduction in the levels of the NOTCH1 transmembrane precursor with the combination of ATRA+DAPT. Overall, the data suggest that the mechanisms underlying N1ICD down-regulation by ATRA and DAPT are different. The results also indicate that ATRA exerts diverse effects on the NOTCH1 pathway in *MB-157* cells relative to the *HCC-1599* and *MDA-MB-157* counterparts. In line with the last observation, ATRA causes a significant down-regulation of the NOTCH1 mRNA in *HCC-1599* and *MDA-MB157* cells, but not in *MB-157* cells (Figure 5B). To evaluate the specificity of the effects exerted by ATRA and DAPT we performed similar experiments in *MDA-MB-231*, *MDA-MB-436* and *HCC-38*, three *TNBC* cell lines of the panel which express significant amounts of the intact NOTCH1 protein (Figure 5C) and do not respond to ATRA (Figure 1 and Figure 3C) or DAPT (Figure 3C). In these cell lines, DAPT and ATRA alone or in combination exert no significant effects on the constitutive amounts of NOTCH1 protein and do not induce the appearance of the N1ICD cleavage product.

### 2.6. Transcriptomic Perturbations Afforded by ATRA and DAPT Alone or in Combination

To get insights into the perturbations afforded by the retinoid and the γ-secretase inhibitor on the gene-transcription profiles of *HCC-1599* and *MB-157* cells, we performed comparative *RNA-seq* studies in the two cell lines exposed to ATRA, DAPT and ATRA+DAPT for 8 h. This time point was selected because we were particularly interested in the early transcriptional changes afforded by the various treatments. In *HCC-1599* cells, ATRA and DAPT up- and down-regulate a large number of genes (FDR < 0.05) (Appendix A; Figure 6A). GSEA (Gene set-Enrichment-Analysis; HALLMARK annotations) of the genes down-regulated by ATRA and DAPT indicates that both compounds modulate the MYC-dependent gene network negatively (Appendix A; Figure 6B). This suggests that down-regulation of the MYC-pathway contributes to the anti-proliferative action of the two compounds. Furthermore, both ATRA and DAPT reduce oxidative phosphorylation, which is consistent with a growth-inhibitory action involving a decrease in mitochondrial activity [36]. As for the up-regulated pathways, ATRA causes a significant enrichment of the “*Interferon-alpha-response*” and “*Interferon-gamma-response*” gene sets [19]. Remarkably, “*Interferon-alpha-response*” is the only up-regulated gene network equally enriched by ATRA and DAPT (Appendix A; Figure 6B). The *RNA-seq* data obtained in *HCC-1599* cells were further analyzed to identify genes commonly up- or down-regulated by ATRA and DAPT. For this purpose, we first reduced the number of potential hits, using a threshold value for the expression fold-change (>40%) caused by ATRA or DAPT (Appendix A; Figure 6A).

ATRA up-regulates 43% and down-regulates 28% of the genes modulated by DAPT in the same direction (Figure 6C,D). The large fraction of common genes regulated by ATRA and DAPT is consistent with a retinoid dependent down-regulation of the N1ICD transcription factor. The ATRA+DAPT combination causes a significantly more sustained up- and down-regulation of 92 (42%) and 23 (38%) of these common genes, respectively (Appendix A; Figure 6A,C,D, genes marked in red). This suggests that many of the common genes are regulated by ATRA and DAPT via different regulatory mechanisms. We used the ATRA/DAPT common genes to generate a NOTCH1-oriented protein-protein interaction network. The results indicate that ATRA and DAPT induce the expression of 23 genes and reduce the expression of 11 genes whose products interact with NOTCH1 directly or indirectly (Appendix A).

In *MB-157* cells, ATRA, DAPT and ATRA+DAPT modulate the expression of a smaller number of genes than in *HCC-1599* cells (Appendix A; Figure 7A). GSEA demonstrates that ATRA up-regulates three gene networks significantly (Figure 7B). As observed in *HCC-1599* cells, “*Interferon-alpha-response*” is one of these top-enriched gene networks. Surprisingly, ATRA up-regulates the MYC-dependent gene network, which is the opposite of what is occurring in ATRA- or DAPT-treated *HCC-1599* cells and DAPT-treated *MB-157* cells. The fraction of genes commonly regulated by ATRA and DAPT (Figure 7C) is much smaller in *MB-157* than *HCC-1599* cells (up-regulated = 29% vs. 42%; down-regulated = 3% vs. 38%). Thus, in *MB-157* cells, ATRA is likely to affect the NOTCH pathway by acting downstream of NOTCH1 expression and N1ICD activation. Co-treatment of *MB-157* cells with ATRA+DAPT enhances the action of ATRA or DAPT in the majority (up-regulated 10/14 = 77%; down-regulated 2/3 = 75%) of these common genes (Appendix A; Figure 7C,D, genes marked in red). This is consistent with the hypothesis that the majority of the ATRA and DAPT common genes are modulated by the two compounds via different mechanisms not only in *HCC-1599*, but also, in *MB-157* cells.

We compared the global effects of ATRA, DAPT and ATRA+DAPT in *HCC-1599* and *MB-157* cells using the 50 HALLMARK gene sets (Figure 7E). Interestingly, ATRA, DAPT and/or ATRA+DAPT down-regulate the “*E2F-targets*”, “*G2M-checkpoint*” and “*MYC-targets*” gene networks in both cell lines. “*E2F-targets*”, “*G2M-checkpoint*” down-regulation may simply be the consequence of the anti-proliferative effect afforded by the two compounds alone or in combination, while the effect on “*MYC-targets*” is likely to be of mechanistic relevance. It is also noticeable that ATRA+DAPT leads to an up-regulation of the “*Interferon-alpha-response*” and “*Interferon-gamma-response*” in both *HCC-1599* and *MB-157* cells.

### 2.7. Role of Rarα and Rarβ in the Anti-proliferative Effects of ATRA

To identify the RAR receptor(s) mediating the activity of ATRA in *TNBC* cell lines, we exposed *HCC-1599*, *MB-157* and *MDA-MB157* cells to AM580 (RARα agonist), UVI2003 (RARβ agonist), BMS961 (RARγ agonist) as well as ATRA for 9 days (Figure 8A). AM580 reduces the growth of these cell lines in a dose-dependent manner and the effects of the RARα agonist and ATRA are quantitatively similar. In contrast, UV2003 and BMS961 do not alter the growth of *HCC-1599*, *MB-157* and *MDA-MB157* cells. The only exception is observed with the highest concentration of BS961 (1 μM). Thus, ligand-dependent activation of RARα seems to be the primary determinant of ATRA-dependent action in *TNBC* cells, as previously observed in the case of luminal breast cancer cell lines [17].

To establish whether ATRA modulates the expression of RARα, RARβ and RARγ, we exposed six *TNBC* and six retinoid-sensitive *Luminal* breast cancer cell lines to ATRA (1 μM) for 24 h (Figure 8B). Consistent with binding, activation and proteasome-degradation of the receptor [37], ATRA reduces the levels of RARα in almost all the cell lines. In contrast, ATRA has no effect on the basal expression levels of RARγ in any cell. As for RARβ, the product of a direct retinoid target-gene [38], no cell line expresses detectable amounts of the receptor in basal conditions. In the *TNBC* context, ATRA up-regulates RARβ only in *HCC-1599*, *MB-157* and *MDA-MB157* cell lines (Figure 8B), indicating an association between RARβ induction and ATRA-sensitivity. In the luminal context, there is no cell line which responds to ATRA with an induction of RARβ.

Thus, RARβ is likely to play a functional role in ATRA growth-inhibitory action only in the case of *TNBC* cell lines. With respect to this, it is remarkable that ATRA does not induce RARβ in DAPT- and retinoid-resistant *MB-157RCL7A* and *MB-157RCL15A* cells (Figure 8C).

To support the functional relevance of RARβ induction, we silenced the retinoid-receptor in *HCC-1599* and *MB-157* cells by stable infection of retroviral constructs containing two distinct shRNAs targeting RARβ (*shRARβ-a* and *shRARβ-b*) and a control shRNA (*shCTRL*). In both *HCC-1599* and *MB-157* cell lines, *shRARβ-a* and *shRARβ-b* suppress the ATRA-dependent induction of RARβ (Figure 8D,E, left). In contrast, *shCTRL* does not alter the up-regulation of RARβ caused by ATRA in parental *HCC-1599* and *MB-157* cells. In these experimental conditions, RARβ knock-down induces ATRA resistance. Indeed, ATRA-dependent growth-inhibition is significantly reduced in *HCC-1599* and *MB-157* cells infected with *shRARβ-a* and *shRARβ-b* relative to the parental or *shCTRL* infected counterparts (Figure 8D,E, right). The functional results obtained indicate that RARβ contributes to the growth-inhibitory action of ATRA in sensitive *TNBC* cell lines.

ATRA-dependent RARβ induction is mediated by RARα activation, as the phenomenon is replicated by AM580 (Figure 8F), while UV2003 and BMS961 are completely inactive. The observation is supported by the results obtained with the RARα antagonist, ER-50891, which blocks the induction of RARβ triggered by ATRA or AM580 in *HCC-1599* cells (Figure 8F). RARβ may contribute to ATRA anti-proliferative action in a ligand-dependent or ligand-independent manner. However, the comparative growth-inhibitory studies performed in *HCC-1599* cells with the above mentioned RAR agonists support the idea that RARβ contributes to ATRA-dependent growth inhibition in a ligand-independent manner (Figure 8G). In fact, the anti-proliferative effect exerted by ATRA in *HCC-1599* cells is entirely recapitulated by the RARα agonist, AM580, which is incapable of binding and activating RARβ at the concentrations used in our experimental conditions.

## 3. Discussion

In previous studies [17,18], we provided pre-clinical evidence that ATRA is particularly effective in *Luminal* and *ER^+^* breast cancer, although a small number of *TNBC* cell lines and primary tumors is also responsive to the retinoid. Here, we profile a large panel of *TNBC* cell lines for their sensitivity to the growth-inhibitory action of ATRA. Among the *TNBC* cell lines considered, only *HCC-1599*, *MB-157* and *MDA-MB157* cells show sensitivity to ATRA. The three cell lines are characterized by specific internal deletions involving the *NOTCH1* gene, indicating that this genetic aberration is a determinant of ATRA sensitivity in *TNBC* cells. In these cells, *NOTCH1* gene internal deletion causes constitutive cleavage of the corresponding protein into the transcriptionally active N1ICD product by γ-secretase [4,23,24,25]. As a consequence, *HCC-1599*, *MB-157* and *MDA-MB157* cells present with high basal levels of NOTCH1-dependent transcriptional activity and require NOTCH1 activation for their growth. NOTCH1 activation renders the three cell lines not only sensitive to ATRA, but also to γ-secretase inhibitors, like DAPT or PF-03084014, as shown in this study, and MRK-003, as demonstrated by Stoeck et al. [24]. In *HCC-1599*, *MB-157* and *MDA-MB157* cells, the sensitivity to ATRA and DAPT or PF-03084014 is quantitatively correlated, suggesting that at least part of the retinoid anti-proliferative effect is due to perturbations of the active NOTCH1 pathway. Indeed, the transcriptomic data obtained in *HCC-1599* and *MB-157* cells indicate that ATRA modulates the expression of various genes which are equally regulated by DAPT-dependent NOTCH1 inhibition.

In *HCC-1599* cells, a large number of genes is commonly regulated by ATRA and DAPT. The phenomenon is partially explained by ATRA-dependent down-regulation of the NOTCH1 mRNA, which causes a decrease in the levels of the constitutively active N1ICD transcription factor. Transcriptional down-regulation of the NOTCH1 gene seems to be also at the basis of the anti-proliferative action exerted by ATRA in the other *MDA-MB157* cell line expressing N1ICD constitutively.

In *MB-157* cells, the mechanisms underlying the effects of ATRA on the NOTCH pathway are different. Consistent with the observation that ATRA does not alter the levels of NOTCH1 mRNA, the number of genes modulated by ATRA and DAPT in a common manner is much smaller in this cell line. Thus, ATRA is likely to target as yet unrecognized elements of the pathway laying downstream of the NOTCH1 protein in *MB-157* cells. Given the role played by NOTCH proteins in the control of Epithelial-to-Mesenchymal-Transition (EMT) [39,40] and the homeostasis of cancer stem cells [41,42], it is likely that ATRA down-modulates both processes [43,44,45]. This is consistent with the differentiating effect of the retinoid and it may result in a reduction of the cancer stem cell component of mammary tumors.

The *RNA-seq* data obtained in *HCC-1599* and *MB-157* cells exposed to ATRA, DAPT and ATRA+DAPT provide clues as to the gene networks and pathways participating in the NOTCH1-dependent anti-tumor action of the two compounds alone or in combination. In *HCC-1599* cells, both ATRA- and DAPT-dependent down-regulation of the NOTCH1 pathway causes a substantial reduction in the expression levels of MYC and the MYC-dependent network. In this cell line, the concordant anti-MYC action of ATRA and DAPT may be one of the major mechanisms underlying the anti-tumor action of the two compounds, as MYC is a well-known regulator of cancer cell stemness [46]. Interestingly, down-regulation of the MYC pathway is enhanced by the combination of ATRA+DAPT, which suggests that the two compounds exert complementary anti-MYC effects. In *MB-157* cells, DAPT causes a similar down-regulation of the MYC gene network, while ATRA treatment results in a surprising up-regulation of the same network. The last observation is consistent with a difference in the mechanisms underlying the anti-tumor action of ATRA in *MB-157* and *HCC-1599* cells, indicating that MYC-targeting is not a major determinant of the *MB-157* growth-inhibitory response to the retinoid. In addition, the *RNA-seq* analyses provide insights as to single genes that may be functionally involved in the anti-proliferative action exerted by ATRA in *HCC-1599* and *MB-157* cells. With respect to this point, the most promising genes are the few ones which are up-or down-regulated in the two cell lines by both ATRA and DAPT. Interestingly, 8 (ELF3, FBXO32, FUT3, GRIN2D, RNASEL, SCNN1A, SREBF1, TINAGL1) of the 220 genes up-regulated by both ATRA and DAPT in *HCC-1599* cells are equally induced by the two compounds in *MB-157* cells. Among the 60 genes down-regulated by the two compounds in *HCC-1599*, ANKRD1 is the only one whose expression is similarly reduced by ATRA and DAPT in *MB-157* cells. All these genes are likely to be functionally involved in the anti-proliferative action exerted by ATRA. Consistent with the fact that the growth-inhibitory action of ATRA and DAPT may be associated with promotion of a differentiating process, ELF3 (ETS-Related-Transcription-Factor-Elf-3) [47], SREBF1 (Sterol-Regulatory-Element-Binding-Protein-1) [48], FUT3 (Galactoside-3(4)-L-Fucosyltransferase) [49], RNASEL (Ribonuclease-L) [50] are involved in fat and epithelial cell-differentiation.

From a basic and mechanistic standpoint, another important result of the study regards the specific involvement of RARβ in the anti-tumor action exerted by ATRA in retinoid-sensitive *TNBC* cells. RARβ is a well-known and direct retinoid-responsive target that can be transcribed from a promoter containing a RAR-binding sequence [12,51,52]. In standard growth conditions, no breast cancer cell line shows detectable levels of the RARβ transcript or protein regardless of the basal or luminal phenotype. Here, we demonstrate that RARβ induction by ATRA is a peculiarity of the retinoid-sensitive *TNBC* cell lines, as indicated by the results obtained in *HCC-1599*, *MDA-MB157* and *MB-157* cells. In fact, a similar induction is not observed in retinoid-resistant *TNBC* cells and any of the luminal cell lines considered, independently of ATRA sensitivity. The reason as to why ATRA up-regulates RARβ only in the *TNBC* cellular context is unknown. However, the phenomenon may be due to regulatory mechanisms involving methylation of the promoter controlling transcription of the RARβ coding gene [53]. Nevertheless, *TNBC*-specific RARβ induction is the consequence of a retinoid-dependent activation of RARα, which is constitutively expressed in all types of breast cancer cells and it is the primary mediator of ATRA activity in mammary tumors [17]. RARβ expression contributes to the anti-proliferative action of ATRA in sensitive *TNBC* cells, as knock-down of the receptor in the *HCC-1599* and *MB-157* cell lines induces partial resistance to the retinoid. It remains to be investigated whether RARβ action is ligand-dependent or -independent, although the data obtained with the specific RARα agonist, AM580, favor the latter hypothesis.

## 4. Materials and Methods

### 4.1. Cell Lines and Chemicals

A list of the characteristics, origin and growth conditions of the cell lines used in the study is available in Appendix A. To obtain DAPT-resistant cell lines, *MB-157* cells were cultured in the presence of the γ-secretase inhibitor (1 μM) for 40 days. Surviving cells were diluted in medium and replated at low density to isolate single-cell derived colonies. These cell cultures were grown in medium containing DAPT (1 μM) for 83 days. Two growing clones were isolated to obtain an equivalent number of DAPT-resistant cell lines (*MB-157RCL7A* and *MB-157RCL15A*). *MB-157* cells stably over-expressing RARβ-targeting shRNAs were obtained by lentiviral infection of constructs based on the pGreenPuro-shRNA system (SBI, System Bioscences). The sequence of the RARβ-targeting shRNAs is available in Appendix A. Lentiviral infected cells were subjected to puromycin (0.5 µg/mL) selection for the isolation of the shRARβ expressing cells.

The sensitivity of cell lines to the anti-proliferative action of ATRA was evaluated with the *ATRA-score* index [17,18]. The following chemicals were used: ATRA (Sigma-Aldrich), AM580 (Tocris), BMS961 (Tocris), UVI2003 (a kind gift of Dr. Angel De Lera, Universidade de Vigo, Spain), ER50891 (Tocris), DAPT (Sigma), PF-03084014 (Pfeizer) and VP16 (SIGMA).

### 4.2. Cell Proliferation Assays and Western Blot Analyses

In the case of adherent cell cultures, cell growth was determined with the use of sulforhodamine assays [54]. In the case suspension cultures, cell growth was evaluated with the CellTiter-Glo Luminescent Cell Viability Assay (Promega), according to the manufacturer instruction. The sensitivity of cell lines to the anti-proliferative action of ATRA was evaluated with the *ATRA-score* index [17,18]. Briefly, breast cancer cell lines were exposed to vehicle (DMSO) or five increasing concentrations of ATRA (0.001–10.0 µM) for 9 days. *ATRA-score* = log2 transformation of the product of AUC X Amax (Area Under the Curve x Maximal Area) rescaled in a range between 0 and 1. “0” and “1” indicate total resistance and maximal sensitivity to ATRA, respectively [18].

Western blot experiments were performed as described [17,34,35]. Briefly, protein lysates from cell cultures were obtained after lysis in RIPA buffer (NaCl 150mM, Igepal 1%, Sodium deoxycholate 0.5%, SDS 0.1%, Tris-HCl, pH 7.5) supplemented with protease inhibitors (Protease Inhibitor Cocktail, Biomak.com and/or PhosSTOP Phosphatase Inhibitor Cocktail, Merck). Nuclear extracts were prepared using the NE-PER™ Nuclear and Cytoplasmic Extraction Reagents according to the manufacturer instruction. Tumor samples obtained from *HCC-1599* xenografts were lysed in NaCl 250mM, Igepal 0.1%, EDTA 5 mM, Tris-HCl (50 mM), pH 7.4 supplemented with protease inhibitors and homogenized mechanically. Cell lysates were separated by SDS-PAGE and finally transferred to nitrocellulose membranes. Membranes were incubated overnight in TBST (NaCl 150 mM, TrisHCl 20mM, Tween 20 0.1%) plus 5% BSA with the following antibodies: anti-RARα, anti-RARβ (provided by Cecile Rochette-Egly, IGBMC, Strasbourg, France), anti-tubulin (Sigma), anti-cleaved-NOTCH1 (V1744) (Cell Signaling), anti-NOTCH1 (Cell Signaling). Blots were subsequently incubated with Cy5-conjugated goat anti-rabbit (GE) or Cy-3 goat anti-mouse (GE) for 1 h at room temperature. Blots were analyzed using an automated fluorescence scanner (Typhoon, GE Healthcare). The densitometric analyses of all the other Western blots presented in the article are available in Appendix A.

### 4.3. RNA Sequencing Studies

RNA sequencing was performed as previously described [18], using the Illumina NextSeq500 apparatus. Alignment of high-throughput, stranded, paired-end reads (151 + 151bp) to the reference genome (hg38), was performed using STAR-aligner, adopting the v27 release of Gencode annotations [55]. Downstream analyses were carried out in the R-statistical-environment, as detailed in Appendix A. The *RNA-seq* data are deposited in the EMBL-EBI Arrayexpress database (Accession No: E-MTAB-9203).

### 4.4. Determination of the NOTCH1 Gene Internal Deletions

Exponentially growing *HCC-1599*, *MB-157* and *MDA-MB157* cells were harvested and RNA extracted using the RNeasy Mini Kit (QIAGEN). RNA was subjected to reverse transcription according to the GeneAmp^®^ RNA PCR Core Kit (Applied Biosystems). The NOTCH1 cDNA was amplified by PCR using primers corresponding to exon 1 and exon 28 (sense: 5′-CCTGCTCTGCCTGGCGCTG-3′; antisense: 5′-CCACGAAGAACAGAAGCACA-3′). The PCR products were purified by gel electrophoresis and subjected to sequencing with the Sanger method (Eurofins Genomics srl) to identify the NOTCH1 internal deletion breakpoint.

### 4.5. Luciferase and Quantitative PCR Assays

Cell lines were co-transfected with the reporter plasmid pGL2HES-1/LUC (HES1-Luc) and the pRL-TK renilla luciferase control reporter construct. Eighteen hours after the transfection, cells were treated with DAPT (1 µM) for 24 h. Luciferase activity was measured with the Dual-Luciferase^®^ Reporter Assay System (Promega) on cell lysates. The expression of the NOTCH1 mRNA was evaluated with the use of a SYBR green assay (Thermo Fisher Scientific) according to the instructions of the manufacturer (Forward primer 5′CCTGCTCTGCCTGGCGCTG3′, exon 1 NOTCH1-gene, nucleotides 283-301 of NM_017617.5; Reverse primer 5′CCACGAAGAACAGAAGCACA3′, exon 28 NOTCH1-gene, reverse strand of nucleotides 5494-5513 of NM_017617.5).

### 4.6. In Vivo Studies

The experimental procedures involving animals were carried out according to the Italian legislation and the Declaration of Helsinki; the studies were approved by the internal Ethical Committee on Animal Experimentation. *HCC-1599* cells (1 × 10^7^/animal) were injected subcutaneously on two flanks of female SCID mice. Four days after transplantation, 12 animals/experimental group were treated with: (a) vehicle (b) PF-03084014 (90 mg/kg, twice daily per os) 5 days a week for a total of 18 days; (c) ATRA (15.0 mg/kg, once/day intraperitoneally) 5 days a week for a total of 18 days; (d) ATRA + PF-03084014 as in (b) and (c). The composition of the vehicle used for ATRA administration was: cremophor EL/ethanol/saline; 0.5/0.5/0.9. The composition of the vehicle used for PF-03084014 administration was: 0.5% methyl cellulose. The administered dosages of ATRA were selected on the basis of previous studies [17,32]. The tumor volume (TV) was determined with a caliper. The growth curves of the tumor xenografts were calculated following normalization for to the TV of the same mouse at the start of treatment. Treatment efficacy was evaluated from the normalized TV curves using two independent parameters: 1) tumor “growth inhibition” (%GI); 2) absolute growth delay (AGD). The %GI value is indicative of short-term antiproliferative effects and it measures the relative tumor growth between the start (day 0) and the end (day 18) of the treatment. The %GI value is calculated adapting the NCI definition according the following mathematical formulae:%GI 0–18 = [(TV18 − 1) / (<TV18> − 1)] × 100; when TV18 ≥ 1%GI 0–18 = [(TV18 − 1)] × 100; when TV18 < 1

In the above formulae, (TV18 − 1) is the increment of the relative tumor weight between day 0 and day 18 of the tumor under analysis, (<TV18> − 1) is the same increment as averages in the control group. AGD, indicative of the long-term delay of tumor regrowth, was calculated as the difference (in days) between the time to reach a target size (four times the size at the start of treatment) in a treated tumor and the mean time to reach the same size in the control group.

## 5. Conclusions

The present study has significant ramifications from both a translation and therapeutic standpoint. In fact, we identify a subset of *TNBC*s characterized by specific genetic aberrations of the NOTCH1 gene which render the tumors markedly sensitive to the anti-proliferative action of ATRA and γ-secretase inhibitors. In addition, we demonstrate a significant cross-talk between the NOTCH and the retinoid signaling pathways which is at the basis of the synergistic growth-inhibitory effects observed with combinations of ATRA and the above mentioned γ-secretase inhibitors. The demonstrated synergistic interactions between ATRA and DAPT or PF-03084014 are of interest for the personalized treatment of this type of mammary tumor. Indeed, combinations of ATRA and γ-secretase inhibitors may represent rational strategies for the personalized treatment of this subset of *TNBC* patients, as they are likely to decrease the systemic toxicity of the latter compounds, which is major clinical problem. In conclusion, the present study represents the basis for the design and conduction of clinical trials aimed at evaluating the efficacy of combinations between ATRA and γ-secretase inhibitors in the treatment of patients affected by *TNBC*s characterized by constitutive activation of the NOTCH1 pathway.

## Figures and Tables

**Figure 1 cancers-12-03027-f001:**
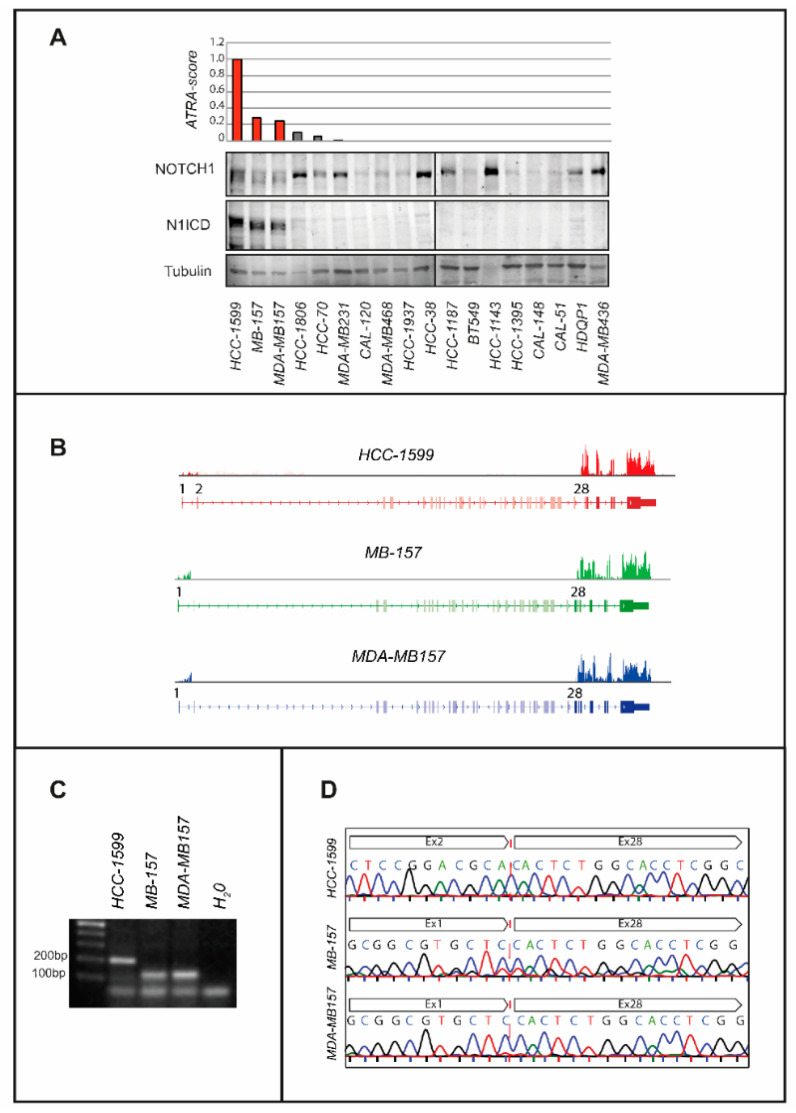
ATRA sensitivity and N1ICD expression in TNBC cell lines. (**A**) Upper: *TNBC* cell lines are ranked in descending order according to the *ATRA-score* shown by the bar graph. The cell lines characterized by internal deletions of the *NOTCH1* gene are marked in red. Each *ATRA-score* value is representative of at least two independent experiments. The cell lines are ordered according to decreasing ATRA-sensitivity from left to right. Lower: The Western blots shown illustrate the levels of NOTCH1, the N1ICD cleavage and transcriptionally active product (Val1744) of NOTCH1 and tubulin (internal control), which were determined using specific antibodies. The densitometric analyses of the Western blots presented are available in Appendix A. (**B**) The panel shows density-plots illustrating the *NOTCH1* gene exon expression imbalances observed in the indicated cell lines. Exons 1-34 of the *NOTCH1* gene are shown. The distribution of the mapped reads aligned to the *NOTCH1* gene is presented as reads x Kilobase x million-reads (RPKM). (**C**) The indicated and exponentially growing cell lines were harvested and subjected to RNA extraction. Following reverse transcription, the cDNA obtained was amplified by PCR using primers corresponding to exon 1 and exon 28 of the *NOTCH1* gene. Fusion amplicons of the expected size were detected. (**D**) The panel shows the sequences of the PCR products obtained in (**C**) and illustrates the fusion breaking points.

**Figure 2 cancers-12-03027-f002:**
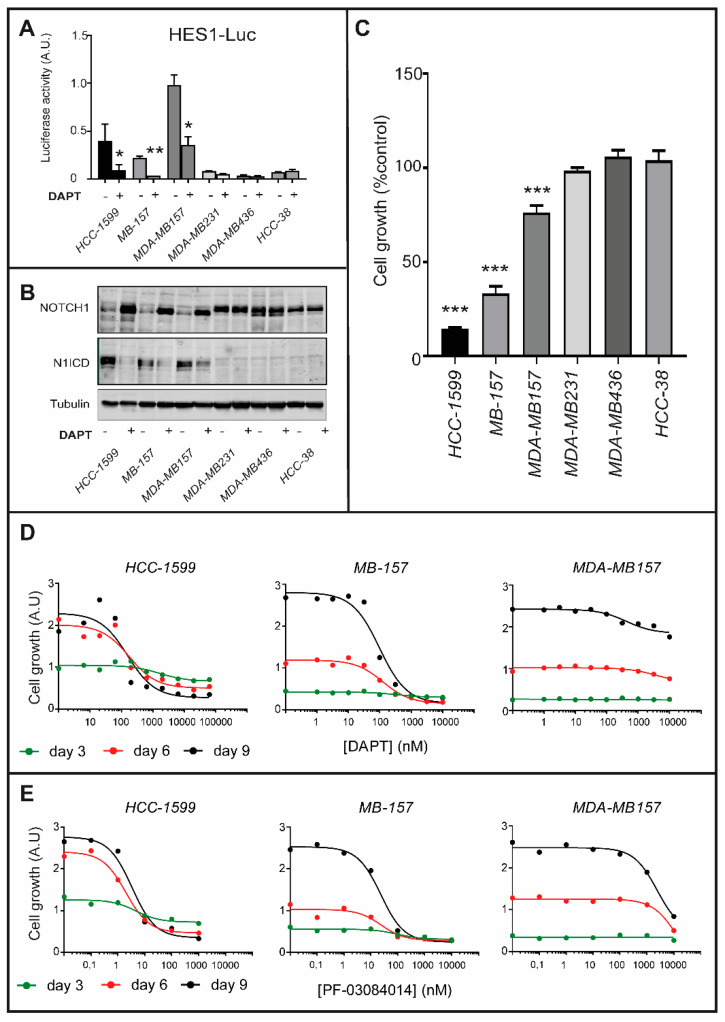
NOTCH1 signaling in ATRA-sensitive/ATRA-resistant TNBC cell lines and role in cellular growth. (**A**) The indicated cell lines were transfected with the reporter plasmid pGL2HES-1/LUC (*HES1-Luc*). The *HES1-Luc* plasmid construct contains the luciferase cDNA under the control of the minimal promoter regulating the expression of the human NOTCH1 target-gene, *HES1*. Eighteen hours post-transfection, cells were treated with DAPT (1 µM) for 24 h. Cell lysates were collected and analyzed for luciferase activity. The *pRL-TK* renilla luciferase plasmid was transfected and used as an internal control to normalize the results for the transfection efficiency. The values are expressed as the Mean + S.D. of 2 independent cultures. The data obtained in each cell line are representative of at least three independent experiments which provided similar results. * Significantly lower relative to the corresponding vehicle treated control (*p* < 0.05, Student’s *t*-test). ** Significantly lower relative to the corresponding vehicle treated control (*p* < 0.01, Student’s *t*-test). (**B**) The indicated cell lines were treated with DAPT (1 µM) for 24 h. Cell lysates were collected and analyzed for NOTCH1, N1ICD or tubulin expression by Western Blot analysis using specific antibodies recognizing each protein. The densitometric analyses of the Western blots presented are available in Appendix A. (**C**) Twenty-four hours following seeding, the indicated cell lines were treated with vehicle DMSO) or DAPT (1 µM) for nine days. Cell growth was determined with the use of the sulforhodamine assay (adherent cell-cultures, *MB-157* and *MDA-MB-157*) or the CellTiter-Glo-Luminescent-Cell-Viability assay (suspension cell-cultures, *HCC-1599*). Cell growth is expressed as the % of the value determined in the various cell cultures exposed to vehicle (CONTROL). Each value is the mean + SD of 6 replicate cultures. The effect of DAPT in the indicated cell lines as compared to *HCC-38* cells (negative control) was determined using Two-way ANOVA followed by a Dunnett’s test (Prism Path 8). *** Significantly different (*p* < 0.001). (**D**,**E**) The indicated cell lines were treated with increasing concentration of the γ-secretase inhibitors, DAPT (**D**) or PF-03084014 (**E**) for 3, 6 and 9 days. Cell growth was determined with the use of the sulforhodamine assay (adherent cell-cultures, *MB-157* and *MDA-MB-157*) or the CellTiter-Glo-Luminescent-Cell-Viability assay (suspension cell-cultures, *HCC-1599*). The panels illustrate the dose-response curves obtained with the PRISM software. Each value is the mean of 6 replicate cultures.

**Figure 3 cancers-12-03027-f003:**
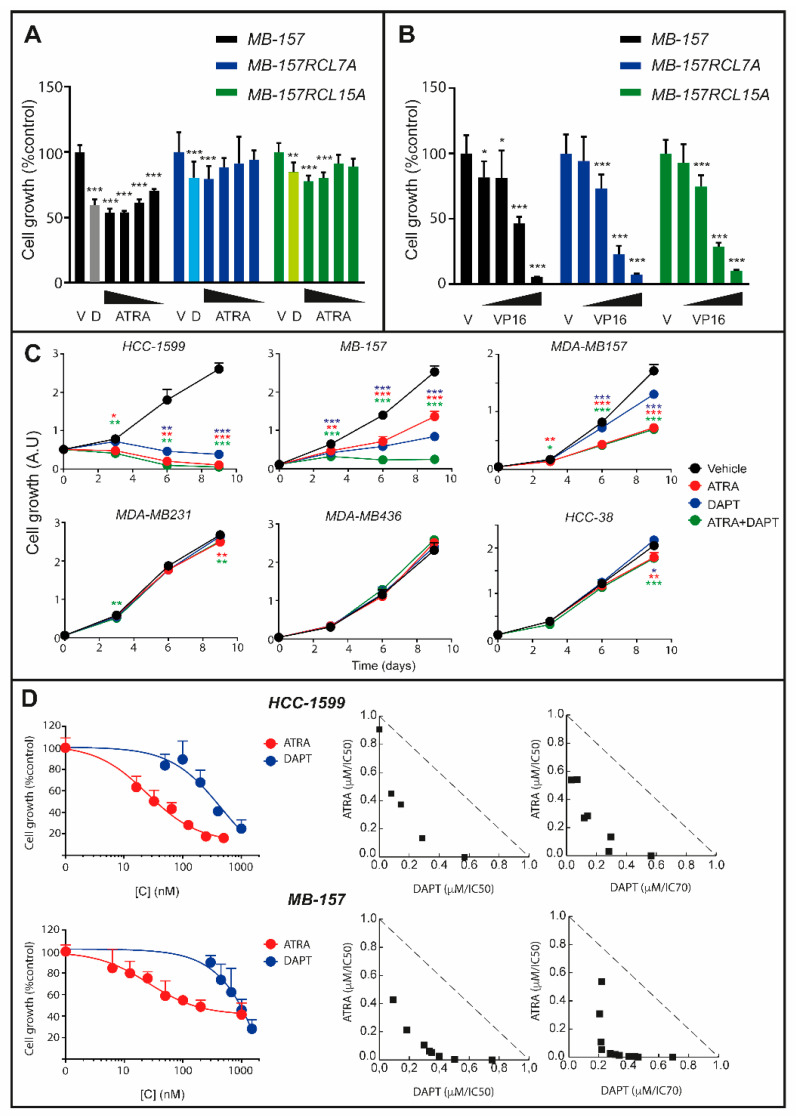
Effects of ATRA and DAPT alone or in combination on the growth of selected TNBC cell lines. (**A**) Parental *MB-157* and DAPT-resistant *MB-157RCL7* or *MB-157RCL15* cells were treated with vehicle (DMSO), DAPT (1 µM) and decreasing concentrations of ATRA (1.0 μM, 0.3 μM, 0.1 μM and 0.03 μM) for 6 days. Cell growth was determined with the sulforhodamine assay. Each column represents the Mean ± SD of 6 replicate cultures. Two-way ANOVA followed by a Dunnett’s test (Prism Path 8) was used for the comparison of each treatment to the corresponding vehicle treated control. *** Significantly different (*p* < 0.001); ** Significantly different (*p* < 0.01). The anti-proliferative effect of DAPT in *MB-157* parental cells (grey column) is significantly higher than in resistant *MB-157RCL7* (light blue column) and *MB-157RCL15* (light green column) cells (*p* < 0.01, Dunnett’s test, Prism Path 8). (**B**) Parental *MB-157* and DAPT-resistant *MB-157RCL7* and *MB-157RCL15* cells were treated with decreasing concentrations (16.0 μM, 3.2 μM, 0.6 μM and 0.1 μM) of VP16 for 3 days. Cell growth was determined with the sulforhodamine assay. Each column represents the Mean ± SD of 6 replicate cultures. Two-way ANOVA followed by a Dunnett’s test (Prism Path 8) was used for the comparison of each concentration of VP16 to the corresponding vehicle treated control. *** Significantly different (*p* < 0.001); * Significantly different (*p* < 0.05). (**C**) The indicated cell lines were treated with ATRA (1 µM), DAPT (1 µM) or the ATRA+DAPT combination for 3, 6 and 9 days, as indicated. Cell growth was determined with the Sulforhodamine assay (*MDA-MB157* and *MB-157* cells) or the CellTiter-Glo-Luminescent-Cell-Viability assay (*HCC-1599* cells). A.U. = Arbitrary Units. Each value is the mean + SD of six independent cultures. The effect of ATRA and DAPT relative to vehicle treated cells was calculated by Two-way ANOVA followed by a Dunnett’s test (Prism Path 8). *** Significantly different (*p* < 0.001); ** Significantly different (*p* < 0.01). * Significantly different (*p* < 0.05). (**D**) *HCC-1599* and *MB-157* cells were treated with vehicle (DMSO) or increasing concentrations of ATRA and DAPT alone or in combination for 9 days. Left: the panels illustrate the ATRA and DAPT dose-response curves obtained on the growth of *HCC-1599* and *MB-157* cells. Each value is the mean + SD of 6 replicate cultures. Right: the panels illustrate the isobolograms of the data obtained with combinations of ATRA and DAPT. The additivity dashed lines separate the antagonistic (upper) from the synergistic (lower) regions.

**Figure 4 cancers-12-03027-f004:**
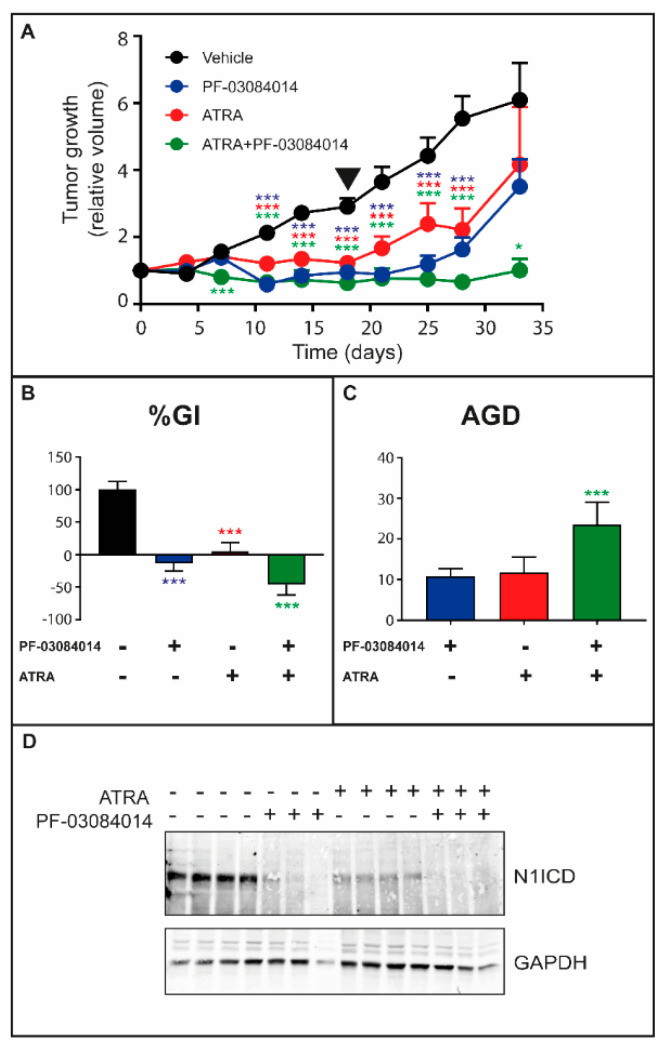
Anti-tumor effects of ATRA and PF-03084014 alone or in combination on the growth of HCC-1599 xenografts. *HCC-1599* cells (1 × 10^7^/animal) were injected subcutaneously on the two flanks of female *SCID* mice. Four days after transplantation 12 animals/experimental group were treated with: (1) vehicle; (2) PF-03084014 (90 mg/kg, *per os* twice/day) 5 days a week for a total of 18 days; (3) ATRA (15.0 mg/kg, intra-peritoneally once/day) 5 days a week for a total of 18 days; (4) ATRA + PF-03084014 as in (2) and (3). The tumor volume (TV) was determined with a caliper. (**A**) The panel shows the growth curves of the *HCC-1599* tumor xenografts. TV values are normalized for the TV measured in the same mouse at the start of the treatment. The triangle indicates the end of the treatment. The effect of ATRA, PF-03084014 and ATRA + PF-03084014 relative to vehicle treated animals was calculated by Two-way ANOVA followed by a Dunnett’s test (Prism Path 8). *** Significantly different (*p* < 0.001); * Significantly different (*p* < 0.05). (**B**,**C**) Treatment efficacy is calculated from the normalized TV curves of the individual mice using two independent parameters: “*Percentage-Growth-Inhibition*” (*%GI*) (**B**) calculated at day 18 (end of the treatment) and “*Absolute-Growth-Delay*” (*AGD*) (**C**). As for *%GI*, each value is the mean+SD of 12 animals. As for *AGD*, each value is the mean + SD of at least 6 animals. One-way ANOVA following Tukey’s Multiple Comparison test (Prism Path 8) was used for the comparison of each treatment to vehicle treated control mice (*** Significantly different (*p* < 0.001). As far as the *%GI* shown in (**B**), Tukey’s Multiple Comparison test reveals a higher efficacy of ATRA + PF-03084014 vs. ATRA (*p* = 0.0002) and ATRA + PF-03084014 vs. PF-03084014 (*p* = 0.0082). (**D**) At the end of the treatment several animals per each experimental group were sacrificed and tumor tissues were isolated. Tissue homogenates were subjected to Western blot analysis with anti-N1ICD and anti-GAPDH antibodies. Each lane corresponds to the lysate derived from a single animal. The densitometric analyses of the Western blots presented are available in Appendix A.

**Figure 5 cancers-12-03027-f005:**
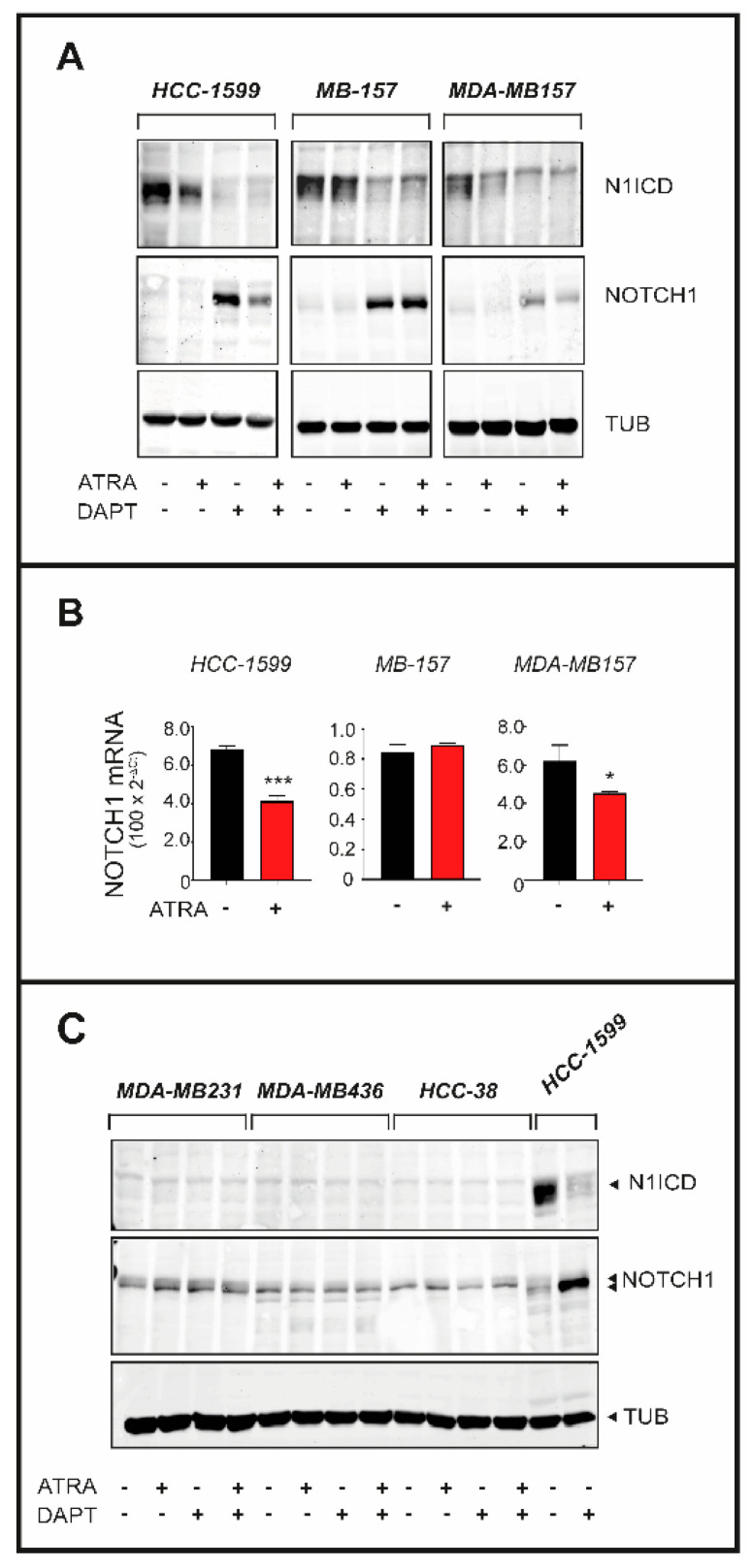
Effects of ATRA and DAPT on the expression of N1ICD and NOTCH1 in retinoid sensitive and retinoid insensitive TNBC cell lines (**A**) HCC-1599, MB-157 and MDA-MB157 cells were treated with vehicle (DMSO), ATRA (1 µM), DAPT (1 µM) or the combination of the two compounds for 24 h. Whole cell lysates were subjected to Western blot analysis for the detection of the NOTCH1 protein and the γ-secretase cleaved N1ICD product using specific antibodies. The same amount of protein was loaded in each lane and the filters were re-blotted with anti-tubulin antibodies, as indicated. The densitometric analyses of the Western blots presented are available in Appendix A. (**B**) RNA extracted from the indicated cell lines exposed to vehicle (DMSO) or ATRA (1 µM) for 24 h was subjected to RT-PCR analysis using a quantitative SYBR green assay for the detection of the NOTCH1 mRNA. The results are the mean + SD of three replicate cell cultures. *** Significantly different (*p* < 0.01, Student’s *t*-test). * Significantly different (*p* < 0.05, Student’s *t*-test). (**C**) MDA-MB231, MDA-MB436 and HCC-38 cells were treated with vehicle (DMSO), ATRA (1 µM), DAPT (1 µM) or the combination of the two compounds for 24 h. Lysates were subjected to Western blot analysis for the detection of the NOTCH1 protein and the γ-secretase cleaved N1ICD product using specific antibodies as in (**A**). HCC1-599 cells were used as an internal control of the experiment. The densitometric analyses of the Western blots presented are available in Appendix A.

**Figure 6 cancers-12-03027-f006:**
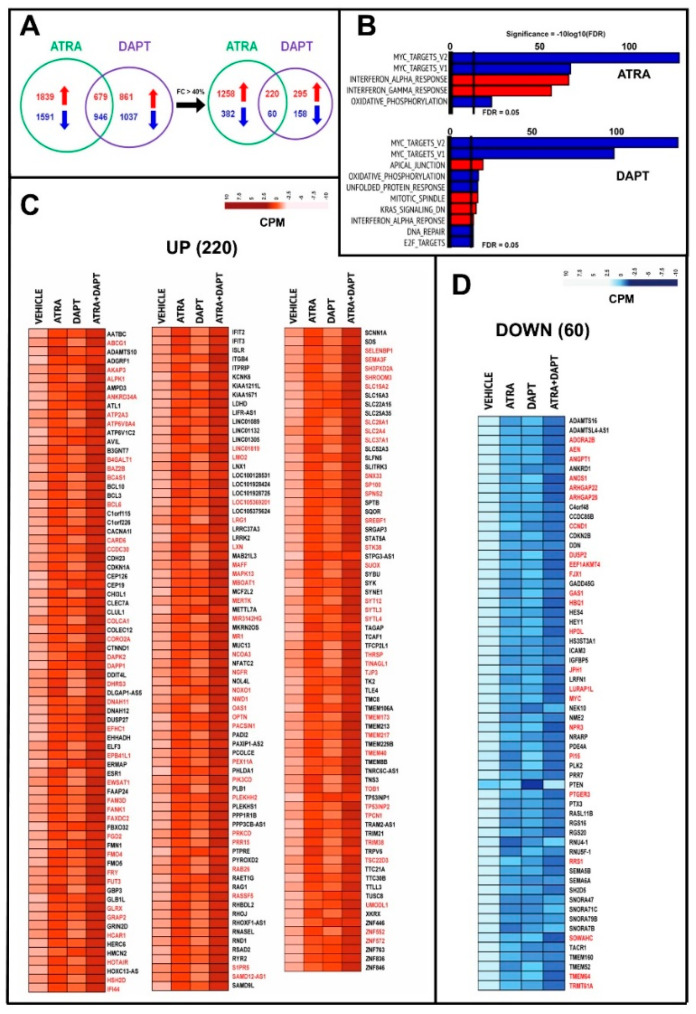
RNA-seq analysis of HCC-1599 cells exposed to ATRA, DAPT and ATRA+DAPT. Three paired biological replicates of *HCC-1599* cells were grown in DMEMF12 medium containing 5% charcolated FBS (Fetal Bovine Serum, Gibco) for 24 h. Cells were treated with vehicle (DMSO), ATRA (1 μM), DAPT (1 μM) or the combination of ATRA and DAPT (ATRA+DAPT) for another 8 h. RNA was extracted and subjected to *RNA-seq* analysis. (**A**) The Venn diagram on the left illustrates the number of genes significantly up-regulated (red) and down-regulated (blue) by ATRA and DAPT. The Venn diagram on the right shows the number of genes significantly up-regulated (red) and down-regulated (blue) by ATRA and DAPT following application of the indicated threshold value (see also Appendix A). (**B**) The panels show the top Hallmark pathways significantly enriched for genes up-regulated (red) or down-regulated (blue) by ATRA and DAPT in *HCC-1599* cells. The vertical black lines indicate the FDR (False Discovery Rate, corrected *p*-value <0.05) thresholds considered. (**C**,**D**) The heat-maps show the expression profiles of the 220 genes commonly and significantly up-regulated (**C**) and the 60 genes commonly and significantly down-regulated (**D**) by both ATRA and DAPT following application of the threshold value indicated in panel (**A**). The effects of the ATRA+DAPT combination are also shown. The up- and down-regulation of the genes marked in red (up-regulated = 92; down-regulated = 23) is significantly enhanced by ATRA+DAPT relative to ATRA and DAPT alone (at least *p* < 0.05 following two-way ANOVA).

**Figure 7 cancers-12-03027-f007:**
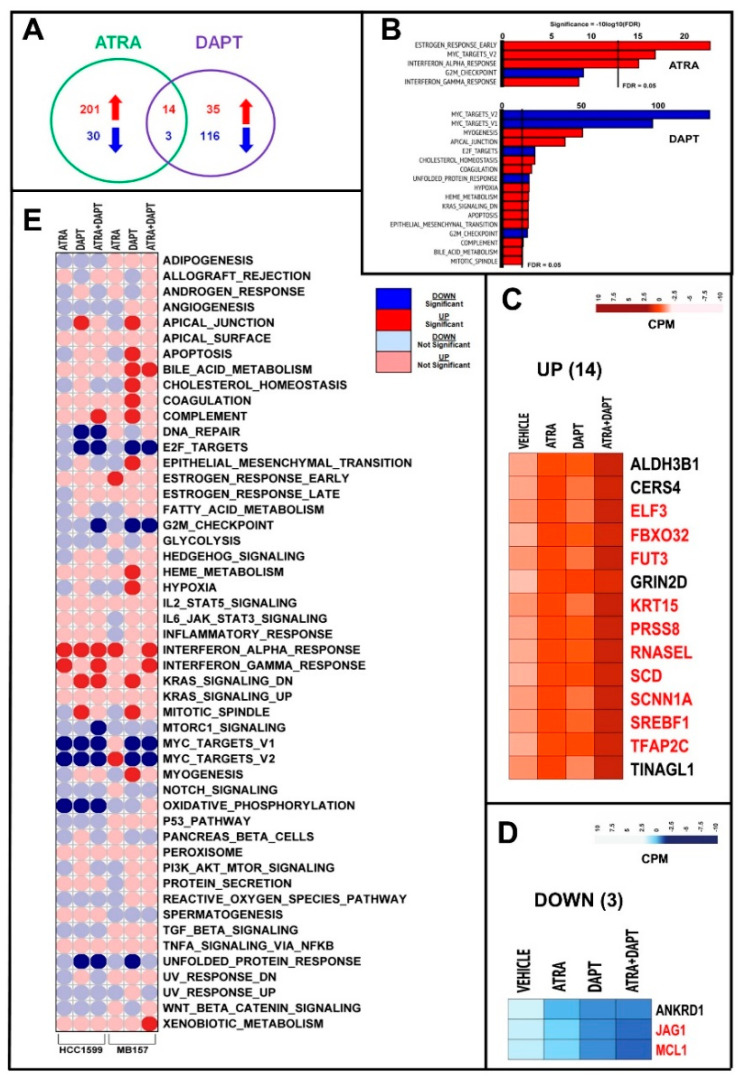
RNA-seq analysis of MB-157 cells exposed to ATRA, DAPT and ATRA+DAPT. Three paired biological replicates of *MB-157* cells were grown in DMEMF12 medium containing 5% charcolated FBS (Fetal Bovine Serum, Gibco) for 24 h. Cells were treated with vehicle (DMSO), ATRA (1 μM), DAPT (1 μM) or the combination of ATRA and DAPT (ATRA+DAPT) for another 8 h. RNA was extracted and subjected to *RNA-seq* analysis. (**A**) The Venn diagram illustrates the number of genes significantly up-regulated (red) and down-regulated (blue) by ATRA and DAPT (see also Appendix A). (**B**) The panels show the top Hallmark pathways significantly enriched for genes up-regulated (red) or down-regulated (blue) by ATRA and DAPT in *MB-157* cells, as indicated. The vertical black lines indicate the FDR (False Discovery Rate, corrected *p*-value < 0.05) threshold considered. (**C**,**D**) The heat-maps show the expression profiles of the 14 genes commonly and significantly up-regulated (**C**) and the three genes commonly and significantly down-regulated (**D**) by both ATRA and DAPT. The effects of the ATRA+DAPT combination are also shown. The up- and down-regulation of the genes marked in red is significantly enhanced by ATRA+DAPT relative to ATRA and DAPT alone (at least *p* < 0.05 following two-way ANOVA). (**E**) Corr-plot of the enrichment results obtained in *HCC-1599* and *MB-157* cells exposed to ATRA, DAPT and ATRA+DAPT considering the 50 Hallmark gene sets. The analysis is based on the following comparisons: ATRA vs. vehicle; DAPT vs. vehicle and ATRA+DAPT vs. vehicle. The red and blue circles represent the Hallmark gene sets up- and down-regulated, respectively. The dark red and dark blue circles indicate the Hallmark gene sets significantly up- and down-regulated with an FDR value < 0.05.

**Figure 8 cancers-12-03027-f008:**
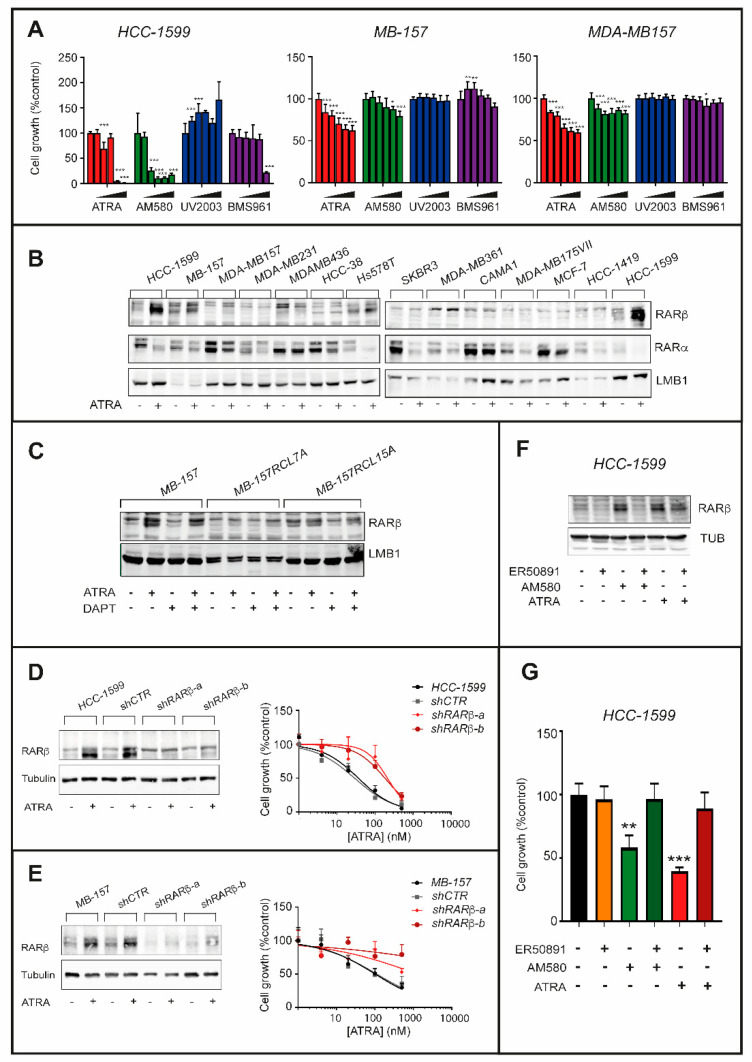
Involvement of RARβ in the anti-proliferative action exerted by ATRA in HCC-1599 and MB-157 cell lines. (**A**) The indicated cell lines were challenged with vehicle (DMSO) and increasing concentrations of ATRA (1.0 μM, 0.3 μM, 0.1 μM, 0.03 μM and 0.01 μM), the RARα agonist, AM580 (1.0 μM, 0.3 μM, 0.1 μM, 0.03 μM and 0.01 μM), the RARβ agonist, UVI2003 (1.0 μM, 0.3 μM, 0.1 μM, 0.03 μM and 0.01 μM), as well as the RARγ agonist, BMS961 (1.0 μM, 0.3 μM, 0.1 μM, 0.03 μM and 0.01 μM), for 9 days. The anti-proliferative effects of each compound were determined and are illustrated by the bar graphs. The values are expressed as the percentage of the sulforhodamine or the CellTiter-Glo-Luminescent-Cell-Viability (*HCC-1599*) results determined in samples treated with vehicle (DMSO) taken as 100. Each value is the Mean + SD of three replicates for *HCC-1599* and 6 replicates for the other cell lines. Two-way ANOVA analysis followed by a Dunnett’s test (Prism Path 8) was used for the comparisons of each dose of the agonist to its relative control (DMSO). The asterisks indicate the *p*-value of each comparison (* *p* < 0.05; ** *p* < 0.01; *** *p* < 0.001). (**B**) The indicated cell lines were challenged with ATRA (1 µM) for 24h. Nuclear extracts were subjected to Western blot analysis for the detection of RARα and RARβ as well as LaminB1 (LMB1, loading control), using specific antibodies. (**C**) Parental *MB-157* as well as DAPT-resistant *MB-157RCL7* and *MB-157RCL15* cells were treated with vehicle (DMSO), DAPT (1 µM), ATRA (1 µM) and the ATRA+DAPT combination for 24 h. Nuclear extracts were subjected to Western blot analysis for the detection of RARα, RARβ and LMB1 (loading control), as in (**B**). The densitometric analyses of the Western blots presented are available in Appendix A. (**D**,**E**) *HCC-1599* and *MB-157* cells were stably infected with lentiviral constructs expressing two distinct shRNAs against RARβ (*shRARβ-a* and *shRARβ-b*) or the empty vector (shCTR). Left panels: cell extracts were subjected to Western blot analysis for the detection of RARβ and Tubulin (loading control) using specific antibodies. Right panels: *HCC-1599* and *MB-157* cells were treated with increasing concentrations of ATRA (4 nM, 20 nM, 100 nM and 500 nM) for 6 days. The anti-proliferative effects of ATRA were determined with the sulforhodamine (*MB-157*) or CellTiter-Glo-Luminescent-Cell-Viability (*HCC-1599*) assays and are illustrated by the graphs. The values are expressed as the percentage of the results obtained following vehicle treatment. Each value is the Mean + SD of at least three replicates. The densitometric analyses of the Western blots presented are available in Appendix A. (**F**) *HCC-1599* cells were challenged with ATRA (60 nM) and the RARα agonist, AM580 (60 nM), in the presence or absence of the RARα antagonist ER50891(1 µM) for 24 h. Cell extracts were subjected to Western blot analysis for the detection of RARα and RARβ proteins as well as Tubulin (loading control), using specific antibodies. The densitometric analyses of the Western blots presented are available in Appendix A. (**G**) *HCC-1599* cells were treated as in (**F**) for 6 days. The anti-proliferative effects of each treatment were determined with the CellTiter-Glo-Luminescent-Cell-Viability assay and are illustrated by the bar graphs. The values are expressed as the percentage of the results determined in samples treated with vehicle (control) taken as 100. Each value is the Mean + SD of three replicates. The effect of each treatment as compared to cells treated with DMSO was calculated by One-way ANOVA followed by Sidak’s test (Prism Path 8). The asterisk indicates the *p*-value of each comparison (*** *p* < 0.001; ** *p* < 0.01).

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
