# Peer review of "Retinoic Acid Sensitivity of Triple-Negative Breast Cancer Cells Characterized by Constitutive Activation of the notch1 Pathway: The Role of Rarβ"

_cancers, 2020, doi:10.3390/cancers12103027_

Round 1

Reviewer 1 Report

Paroni et al. provide evidence on the potential of retinoid acid (ATRA) for the treatment of certain sub-populations of Triple Negative Breast Cancers. More specifically, the Authors demonstrate that ATRA sensitivity is associated with genetic alterations of the NOTCH1-gene, leading to constitutive activation of the y-secretase product named N1ICD. Interestingly, cell lines with constitutive activation of NOTCH-1 and sensitivity to ATRA are also sensitive to the anti-proliferative action of y-secretase inhibitors. In addition, the combined treatment of ATRA and y-secretase inhibitors induce synergistic effects in cell models as well as in animal models of TNBC. The effects induced by ATRA are mediated by the receptor RARβ. Performing RNA-seq experiments in cell lines treated with ATRA and a y-secretase inhibitor, the Authors demonstrate that the treatments regulate a set of shared genes belonging to the Notch pathway and also regulating an anti-viral immune response. 

The Manuscript is written in a clear and detailed manner. The experimental design adequately supports the conclusions of the Authors. There are few aspects that need to be addressed:

1) In western blotting, densitometric analysis of the blots should be included.

2) The expression of Notch1 and N1ICD in tumor samples obtained from xenografts would corroborate the Authors' findings.

3) There are some typos throughout the manuscript that needs to be corrected.

4) In the Abstract "Triple-negative breast-cancer (TNBC) is a heterogeneous disease and lacks effective therapeutic options", substitute "and" with "that".

5) Sometimes the Figures are not self-explanatory (for instance in Figure 8 panel G it's not immediate we are talking about proliferation assay.

Author Response

Comments for Notch 1 paper form Cancers  08102020

Reviewer 1:  Comments and Suggestions for Authors

Paroni et al. provide evidence on the potential of retinoid acid (ATRA) for the treatment of certain sub-populations of Triple Negative Breast Cancers. More specifically, the Authors demonstrate that ATRA sensitivity is associated with genetic alterations of the NOTCH1-gene, leading to constitutive activation of the y-secretase product named N1ICD. Interestingly, cell lines with constitutive activation of NOTCH-1 and sensitivity to ATRA are also sensitive to the anti-proliferative action of y-secretase inhibitors. In addition, the combined treatment of ATRA and y-secretase inhibitors induce synergistic effects in cell models as well as in animal models of TNBC. The effects induced by ATRA are mediated by the receptor RARβ. Performing RNA-seq experiments in cell lines treated with ATRA and a y-secretase inhibitor, the Authors demonstrate that the treatments regulate a set of shared genes belonging to the Notch pathway and also regulating an anti-viral immune response. 

The Manuscript is written in a clear and detailed manner. The experimental design adequately   supports the conclusions of the Authors. There are few aspects that need to be addressed:

  • In western blotting, densitometric analysis of the blots should be included.

Response:  In the first submission of the manuscript, the densitometric data of the Western blot analyses presented were already available in the Original Figure file.  We still think that addition of the densitometric data makes the figures overcrowded and unclear.  Nevertheless, we do agree with the Reviewer that the densitometric data should be more accessible.  Please notice that we added the Western blot data obtained on xenografts (see response to point 2 of the Reviewer, below) in Figure 4D.   For all these reasons, we included the original versions of the Western blots along with the densitometric values in Supplementary Figures S5 to Supplementary Figures S10. In the Materials and Methods section of the manuscript, we added a new statement “The densitometric analyses of all the other Western blots presented in the article are available in Supplementary Figures S5-S10” to clarify the point (page 18, lines 664-665). In addition, the relevant Supplementary Figures containing the densitometric data are quoted in the legends to Figure 1A (page 4, lines 153-154), Figure 2B (page 7, lines 236-237), Figure 4 (page 9, lines 304-305), Figure 5A and 5C (page 10, lines 359-360; page 11,  lines 369-370), Figure 8B, 8C,  8D, 8E and 8F (page 16, lines 542-543; page 16 lines 551-552; page 16, lines 556-557).     

  • The expression of Notch1 and N1ICD in tumor samples obtained from xenografts would corroborate the Authors' findings.

Response: We thank the Reviewer for the useful suggestion, particularly in consideration of the fact that we conducted studies on the in vivo effects exerted by ATRA and PF-03084014 on N1ICD in HCC-1599 xenografts and we did not present the corresponding data in the original version of the manuscript.  We evaluated the levels of N1ICD in the xenografts of HCC-1599 cells by Western blot analysis.  To perform these studies, we isolated the tumors of 3-4 animals treated with vehicle, ATRA, PF-03084014 and the combination of the two compounds.  The tumor extracts were subjected to Western blot analyses with anti-N1ICD and anti-GAPDH antibodies.  In the new version of the manuscript, the results of these analyses have been added in Figure 4D.  In addition, the densitometric data of the Western blots are presented in Supplementary Figure S7.   The data confirm the results obtained on the same cellular model in vitro and corroborate our findings. The new results are described on pages 5-6 lines 213-220: “Consistent with what is observed in the cultures of HCC-1599 cells, the tumor tissue of vehicle treated control animals expresses only the N1ICD form of the NOTCH1 receptor (see Figure 1A).  In addition and as expected, the antitumor effects observed with PF-03084014 are associated with a substantial reduction in the levels of the N1ICD protein synthesized by the breast cancer cells [31] (Figure 4D).  Interestingly, a similar reduction in the amounts of the N1ICD protein is also observed following administration of ATRA.  Finally, the N1ICD protein is undetectable in tissue samples obtained from animals treated with the ATRA+PF-03084014 combination.”.  Please notice that we modified the legend to Figure 4 to comply with the addition of the new panel 4D (page 8-9, lines 281-305).   

3) There are some typos throughout the manuscript that needs to be corrected.

Response: We revised the entire text including the legends to figure and we corrected all the typos.

4) In the Abstract "Triple-negative breast-cancer (TNBC) is a heterogeneous disease and lacks effective therapeutic options", substitute "and" with "that".

Response:  We corrected the text as suggested by the reviewer (page 1, line 32).

5) Sometimes the Figures are not self-explanatory (for instance in Figure 8 panel G it's not immediate we are talking about proliferation assay.

We do agree with the reviewer that Figure 8G is unclear.  This is due to the fact that the vertical axis lacks the legend.  The new version of Figure 8G contains this legend “Cell Growth (% control)”.  This addition makes the content of the figure clearer and addresses the criticism raised by the Reviewer.

Reviewer 2 Report

Comments for Authors:

Dear Authors,

the studies were properly planned and conducted, and the manuscript correctly written. However, I have a few comments.

Please respond to the comments below:

  1. In the Introduction in lines 59-60 Authors stated “The final γ-secretase-dependent cleavage of NOTCH1 causes the release and nuclear translocation of 59 the receptor intracellular domain (N1ICD), which is an active transcription factor [7, 8]. It seems to me that due to the fact that NOTCH and N1ICD constitute the main issue of this manuscript, it would be necessary to describe, at least briefly, the genes whose expression is affected by N1ICD transcription factor.
  2. Is it known what is the mechanism of proliferation stimulation by N1ICD - the transcriptionally active form of NOTCH1?
  3. Why did you choose to incubate cells with DAPT inhibitor for 9 days in most proliferation assays? It’s quite a long time in this kind of test. Have you changed the cell culture medium in the meantime? Do you know what is the half-life of DAPT?
  4. 2 – In my opinion giving the concentrations of the drugs in the form of log on the charts is a bit misleading because it does not allow the reader to know what were really used (it seems to me that you used conc. in the range of 10µM-10mM, so quite high)
  5. Lines 164-165 – why only two, not three independent cultures were used to perform the experiments?
  6. Lines 169-170 – what was the concentration of DMSO used in a control condition?
  7. Line 170-172 - I am not sure if the data presenting the results of different tests should be included in one graph
  8. In the description of Figure 2 C Authors stated “Cell-growth was determined with the use of the sulforhodamine (adherent cell-cultures, MB-170 157 and MDA-MB-157) assay or the CellTiter-Glo-Luminescent-Cell-Viability assay (suspension cell cultures, HCC-1599), while in the description of Figure 2 D,E it is writen that “Sulforhodamine assays 177 were performed to determine the growth of each cell-line.” Which version is correct?
  9. Lines 185-186 - Typically resistant cells are obtained by incubation of cells with increasing doses of drug/inhibitor. Here, only one concentration of DAPT was used to obtain resistant cells- why?
  10. Figure 3 – why the cell line MB-157 was chosen to obtain resistant cell line? Why in part A cells were incubated with drugs for 6 days, while in part B for 3 days? Why as a chemotherapeutic in part B was chosen VP16?
  11. Lines – 255-256 – In my opinion statement “In our experimental conditions, treatment of the tumor bearing 254 animals ATRA, PF-03084014 or ATRA+PF-03084014 seems devoid of significant systemic toxicity, as 255 indicated by the lack of significant effects on body weight” is too strong. Do you think it is possible to determine if a drug exhibit systemic toxicity just by its effect on the weight of the mice?

Author Response

Reviewer 2:  Please respond to the comments below:

  1. In the Introduction in lines 59-60 Authors stated “The final γ-secretase-dependent cleavage of NOTCH1 causes the release and nuclear translocation of the receptor intracellular domain (N1ICD), which is an active transcription factor [7, 8]. It seems to me that due to the fact that NOTCH and N1ICD constitute the main issue of this manuscript, it would be necessary to describe, at least briefly, the genes whose expression is affected by N1ICD transcription factor.

Response:  As suggested by the reviewer, in the Introduction section, we added a very brief statement regarding some of the target genes whose expression is modulated by N1ICD: “The final γ-secretase-dependent cleavage of NOTCH1 causes the release and nuclear translocation of the receptor intracellular domain (N1ICD), which is part of an active transcriptional factor complex controlling the expression of various target genes [7, 8].  Among the known target genes, members of the HES and HEY families, CyclinD1 and cMyc stand out [3]. Some of these genes, with particular reference to cMyc, are involved in the proliferative effects induced by the activation of the NOTCH pathway in certain types of leukemia and solid tumors. All this supports …” (page 2, lines 60-66). 

  1. Is it known what is the mechanism of proliferation stimulation by N1ICD - the transcriptionally active form of NOTCH1?

Response: The point raised by the Reviewer is the object of numerous studies.  It is clear that the mechanisms underlying the proliferative action of N1ICD are tumor type specific. cMyc is a direct target of N1ICD and it is certainly one of the best characterized genes in terms of its involvement in cancer and leukemia cell growth.  We briefly mentioned the issue in the Introduction of the new version of the manuscript: “Some of these genes, with particular reference to cMyc, are involved in the proliferative effects induced by the activation of the NOTCH pathway in certain types of leukemia and solid tumors. All this supports …”  (page 2, lines 64-66), as also mentioned in the response to point 1 of the Reviewer.

  1. Why did you choose to incubate cells with DAPT inhibitor for 9 days in most proliferation assays? It’s quite a long time in this kind of test. Have you changed the cell culture medium in the meantime? Do you know what is the half-life of DAPT?

Response:  The growth of most of the cell-lines used throughout the study is relatively slow and determination of an anti-proliferative effect requires at least 2-3 replication cycles.  For this reason, we decided to use the 9-day time-frame to perform most of our studies.  We routinely changed the medium every 3 days.  In our experimental conditions, we never measured the half-life of DAPT.  Nevertheless, we checked that the γ-secretase inhibitory activity of DAPT was maintained for at least 3 days.   

  1. 2 – In my opinion giving the concentrations of the drugs in the form of log on the charts is a bit misleading because it does not allow the reader to know what were really used (it seems to me that you used conc. in the range of 10µM-10mM, so quite high)

Response:  To facilitate the readership we changed the scales of the horizontal axis of Figures 2D, 2E, 8D and 8E and Supplementary Figure S1 from a log to a linear form, according to the suggestions of the Reviewer.  However, please notice that with the exception of DAPT (63 mM), we never used concentrations of any drug above 10 mM, as also indicated by the new versions of the Figures.

  1. Lines 164-165 – why only two, not three independent cultures were used to perform the experiments?

Response:  The results of the studies presented in Figure 2A are the mean of two independent cultures and the data obtained in each cell line are representative of at least three independent experiments which provided similar results.  The point is clarified in the new version of the legend to Figure 2A (page 6, lines 230-232).  Please notice that the data obtained simply confirm the γ-secretase inhibitory activity of DAPT, which is widely described in the scientific literature.   

  1. Lines 169-170 – what was the concentration of DMSO used in a control condition?

Response:  The concentration of DMSO used in all the studies never exceeded 0.1% v/v.

  1. Line 170-172 - I am not sure if the data presenting the results of different tests should be included in one graph

Response:  Please notice that the results presented in Figure 2C are expressed as cell growth (% control).  Hence, all the data are already normalized for the vehicle treated control taken as 100% and the results are comparable across the cell-lines.  This is the reason as to why it is our opinion that the Figure should be maintained as it is.  

  1. In the description of Figure 2 C Authors stated “Cell-growth was determined with the use of the sulforhodamine (adherent cell-cultures, MB-170 157 and MDA-MB-157) assay or the CellTiter-Glo-Luminescent-Cell-Viability assay (suspension cell cultures, HCC-1599), while in the description of Figure 2 D,E it is written that “Sulforhodamine assays 177 were performed to determine the growth of each cell-line.” Which version is correct?

Response:  We thank the Reviewer for noticing the mistake present in the legend to Figures 2D and 2E.  We have corrected the mistake in the new version of the legend to Figure 2: “Cell-growth was determined with the use of the sulforhodamine assay (adherent cell-cultures, MB-157 and MDA-MB-157) or the CellTiter-Glo-Luminescent-Cell-Viability assay (suspension cell-cultures, HCC-1599).” (page 7, lines 239-241).

  1. Lines 185-186 - Typically resistant cells are obtained by incubation of cells with increasing doses of drug/inhibitor. Here, only one concentration of DAPT was used to obtain resistant cells- why?

Response:  The concentration of DAPT used to obtain a resistant population of MB-157 cells, is based on a series of pilot experiments that allowed the selection of this optimal concentration.     

  1. Figure 3 – why the cell line MB-157 was chosen to obtain resistant cell line? Why in part A cells were incubated with drugs for 6 days, while in part B for 3 days? Why as a chemotherapeutic in part B was chosen VP16?

Response:  As to the first point, we chose the MB-157 cell line, because: 1) in this cell-line, NOTCH1 is constitutively activated (see Figures 1 and 2);  2) NOTCH1 signaling is required for MB-157 cell proliferation (see Figure 2); 3) unlike HCC-1599, MB-157 cells grow in adherence and can be selected for DAPT resistance much more easily.  As to the second point raised by the Reviewer, in breast cancer, the anti-tumor action of ATRA is predominantly due to anti-proliferative and cyto-differentiating effects (see reference 17 and reference 34).  In particular ATRA is devoid of a direct cytotoxic action.  For this reason, the action of ATRA is relatively slow (see also the response to point 3 of Reviewer 1).  In contrast, the action of classic chemotherapeutics, like VP16, relies on direct cytotoxic and apoptotic effects, which are much faster processes.  This explains the use of two different time-frames in panel A and panel B of Figure 2.  Please, notice that the results obtained with VP16 were confirmed with the use of another classic chemotherapeutic, like paclitaxel.  However, we did not deem it necessary to add the data on paclitaxel, as they do not add further relevant pieces of information.

  1. Lines – 255-256 – In my opinion statement “In our experimental conditions, treatment of the tumor bearing 254 animals ATRA, PF-03084014 or ATRA+PF-03084014 seems devoid of significant systemic toxicity, as 255 indicated by the lack of significant effects on body weight” is too strong. Do you think it is possible to determine if a drug exhibit systemic toxicity just by its effect on the weight of the mice?

Response:  In the new version of the manuscript we eliminated the reference to the lack of systemic toxicity, although we left the results of the body weight measurement: “In our experimental conditions, treatment of the tumor bearing animals with ATRA, PF-03084014 or ATRA+PF-03084014 do not cause significant effects on the body weight of xenografted animals (Supplementary Figure S3).” (page 5, lines 211-213).  Indeed, body weight is one of the standard parameters that is measured in vivo to evaluate animal health and this piece of information should be kept in the manuscript.

Reviewer 3 Report

The manuscript “Retinoic Acid Sensitivity of Triple-Negative Breast Cancer Cells Characterized by Constitutive Activation of the NOTCH1 Pathway: the Role of RARβ” by Paroni et al. describes the effect of combination treatment of all-trans retinoic acid and γ-secretase inhibitors on the subset of triple negative breast cancer cell lines characterized by NOTCH1 aberrations. The manuscript is written with the proper scientific English language, figures are esthetic and high quality, the results support the conclusions, and those are clearly formed. The manuscript should overall meet the interest of the readers of Cancers. Below I am addressing some minor concerns for authors:

  1. Please explain the abbreviation “ATRA” in the abstract.
  2. Line 65, reference is missing, please add.
  3. Figure 1 A tubulin, the loading control is not evenly expressed. Please explain.
  4. Figure 2, please provide the rationale for the use of DAPT in the dose of 1 uM.
  5. In vivo study, please provide the information, what was administered to the mice as a vehicle.
  6. Line 646, the number of cells is 1x107/animal. In the Figure 4 different number is provided: 1x10^7/animal. Please correct.
  7. Please explain how the doses of compounds in in vivo studies were calculated, since in in vitro studies cells were subjected to the treatment with equal does of ATRA and γ-secretase inhibitors.

Author Response

The manuscript “Retinoic Acid Sensitivity of Triple-Negative Breast Cancer Cells Characterized by Constitutive Activation of the NOTCH1 Pathway: the Role of RARβ” by Paroni et al. describes the effect of combination treatment of all-trans retinoic acid and γ-secretase inhibitors on the subset of triple negative breast cancer cell lines characterized by NOTCH1 aberrations. The manuscript is written with the proper scientific English language, figures are esthetic and high quality, the results support the conclusions, and those are clearly formed. The manuscript should overall meet the interest of the readers of Cancers. Below I am addressing some minor concerns for authors:

Please explain the abbreviation “ATRA” in the abstract.

Response: we added “all-trans retinoic acid” in the abstract (page 1, line 34).

Line 65, reference is missing, please add. 

As requested by the Reviewer we added the following reference (ref. 11) : Locatelli M, Curigliano G. Notch inhibitors and their role in the treatment of triple negative breast cancer: promises and failures. Curr Opin Oncol. 2017 Nov;29(6):411-427. doi: 10.1097/CCO.0000000000000406. PMID: 28914645.  We adjusted the order of all the other references accordingly.

Figure 1 A tubulin, the loading control is not evenly expressed. Please explain.

Response: The figure shows the results obtained in different cell-lines which contain different amounts of tubulin.  Nevertheless the quantity of protein loaded in each cell-line is the same as evaluated by the BCA assay and Ponceau staining. 

Figure 2, please provide the rationale for the use of DAPT in the dose of 1 uM.

Response:  The selection of the DAPT concentration is based on pilot experiments aimed at defining the optimal anti-proliferative concentration of the γ-secretase inhibitor across the different cell lines used.

In vivo study, please provide the information, what was administered to the mice as a vehicle.

Response:  The composition of the vehicle used for ATRA administration was: cremophor EL/ethanol/saline; 0.5/0.5/0.9.  The composition of the vehicle used for PF-03084014 administration was: 0.5% methyl cellulose.  This information was added in the Materials and Methods section (page 19, lines 703-705).

Line 646, the number of cells is 1x107/animal. In the Figure 4 different number is provided: 1x10^7/animal. Please correct.

Response:  We thank the reviewer for noticing the mistake in the Materials and Methods section.  We corrected the mistake (page 19,  line 698).  

Please explain how the doses of compounds in in vivo studies were calculated, since in in vitro studies cells were subjected to the treatment with equal does of ATRA and γ-secretase inhibitors.

Response:  The dosage of ATRA administered to tumour bearing mice in vivo is equivalent to the dosage employed in humans for the treatment of acute promyelocytic leukemia.  The same dosage of ATRA has already been used in other studies conducted by our group (see for instance reference [17].  As for the dosage of PF-03084014, the dosage used is based on a study available in the literature (see reference [32]).

Round 2

Reviewer 1 Report

The Authors have addressed all the requests.

In my opinion, the Manuscript is now suitable for publication.